# Timescale of local moment screening
# across and above the Mott transition

Léo Gaspard [1] and Jan M. Tomczak [2*]

**1** Laboratoire de Chimie et Physique Quantiques, Université Paul Sabatier Toulouse III, CNRS,
118 Route de Narbonne, 31062 Toulouse, France
**2** Institute of Solid State Physics, TU Wien, Vienna, Austria * jan.tomczak@tuwien.ac.at

December 7, 2021

## Abstract

A material's phase diagram typically indicates the types of realized long-range orders, corresponding to instabilities in *static* response functions. In correlated systems, however, key phenomena crucially depend on *dynamical* processes, too: In a Mott insulator, the electrons' spin moment fluctuates in time, while it is dynamically screened in Kondo systems. Here, we introduce a *timescale* $t_m$ characteristic for the screening of the local spin moment and demonstrate that it fully characterizes the dynamical mean-field phase diagram of the Hubbard model: The retarded magnetic response delineates the Mott transition and provides a new perspective on its signatures in the supercritical region above. We show that $t_m$ has knowledge of the Widom line and that it can be used to demarcate the Fermi liquid from the bad metal regime. Additionally, it reveals new structures *inside* the Fermi liquid phase: First, we identify a region with preformed local moments that we suggest to have a thermodynamic signature. Second, approaching the Mott transition from weak coupling, we discover a regime in which the spin dynamics becomes adiabatic, in the sense that it is much slower than valence fluctuations. Our findings provide resolution limits for magnetic measurements and may build a bridge to the relaxation dynamics of non-equilibrium states.

# 1  Introduction

Every electron has a spin [1]. How this intrinsic magnetic moment manifests, however, crucially depends on the hosting environment. In a solid, the intuitive picture is as follows: The configuration of open-shell atoms fluctuates around an average valency on a characteristic timescale $t_{hyb}$. Taking a snapshot, a fraction of lattice sites will have their low-energy orbitals *simultaneously* occupied by two electrons of opposite spin (or by no electrons at all). These magnetically inactive sites effectively reduce the system's average *instantaneous* moment: Only sites hosting an unpaired spin can contribute to the local moment. Over time, however, electrons of opposite spin (Pauli principle) can stopover at such sites. This singlet-by-visitation dynamically screens the local moment. To quantify the screening dynamics, we introduce a *characteristic timescale $t_m$* that describes the retarded decay of the local moment. In the weak coupling Pauli regime, $t_m$ is short since it is set by the inverse of the electrons' kinetic energy. Toward stronger coupling, double occupations—penalized by the on-site interaction—diminish, enhancing the instantaneous moment. While it is still screened, the decay is slower, as kinetic energy dwindles. This scenario changes radically when passing through the Mott metal-insulator transition: At strong coupling, magnetic screening is ineffective as charge fluctuations are quenched. While double occupations subside and the instantaneous moment approaches the electron's intrinsic spin moment, it hardly decays over time, resulting in a *persisting* local moment and a Curie susceptibility.

    In this work, we detail and quantify the above intuitive picture for the Hubbard model and answer the following central questions: How does the characteristic timescale $t_m$ of magnetic screening evolve with interaction strength $U$ and temperature $T$? What happens when approaching and passing *through* the Mott metal-insulator coexistence region below its critical endpoint

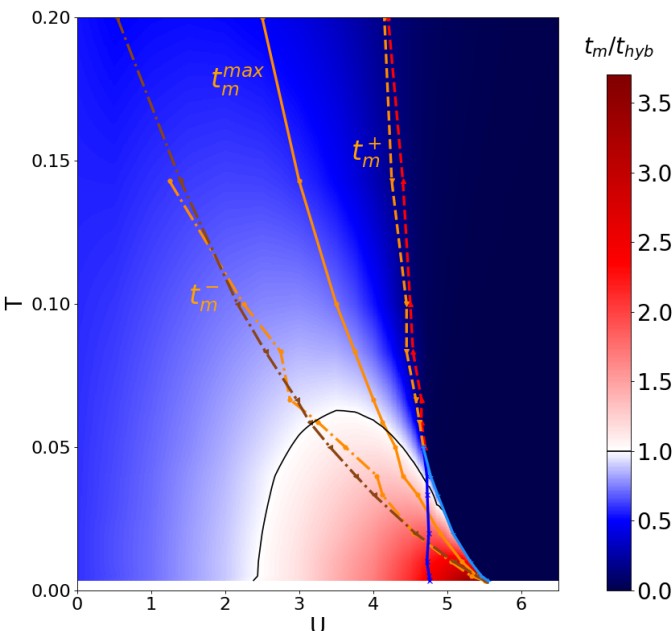

Figure 1: Phase diagram of the Hubbard model according to the timescale ratio $t_m/t_{hyb}$, where $t_m$ ($t_{hyb}$) is the characteristic timescale of the local moment screening (valence fluctuations). $t_m/t_{hyb} > 1$ [< 1] indicates valence fluctuations to be faster [slower] than the screening of the local moment (blue [red]). Close to the Mott transition, an adiabatic spin regime emerges (dark red), where the spin dynamics is much slower than valence fluctuations. Orange lines follow maxima $t_m^{\text{max}}$ (full) and inflection points $t_m^-$ (dashed-dotted), $t_m^+$ (dashed) in $t_m(U)$ at constant $T$. The brown line (dash-dotted) distinguishes a magnetic susceptibility dominated by short-time dynamical contributions (below) from a preponderant incipient local moment (above the line). The Widom line is shown in red (dashed). Blue lines delimit the metal-insulator coexistence region ($U_{c1} < U < U_{c2}$), inside which we display the metallic solution. Results are numerically exact for the $D = \infty$ Bethe lattice. Energies are given in units of the hopping $t$.

and what are the transition's signatures in the spin dynamics *above* it? Our key findings are: The iconic interaction vs. temperature phase diagram [2] can be entirely characterized by anatomizing the magnetic screening timescale $t_m$, see Fig. 1: All transition and crossover lines previously assigned on the basis of spectral properties [3,4] (coexistence region, super-critical crossover), the double occupancy (Widom line [5–7]), or the electrical conductivity [6,8] or the static magnetic susceptibility [9] (Fermi liquid to bad metal crossover), can be extracted and interpreted on the basis of the dynamical screening of the local moment. Beyond previous classifications, $t_m$ reveals additional crossovers within the Fermi liquid phase: First, between a Pauli-like region and a local moment precursor regime which we suggest to be related to a thermodynamic signature in the electronic specific heat [10]. Second, on the metallic side of the Mott transition we identify an extended regime, marked in red in Fig. 1, in which the spin dynamics is *adiabatic* in the sense that magnetic screening is significantly slower than valence fluctuations ($t_m \gg t_{hyb}$). This criterion in particular distinguishes the Fermi liquid realized at intermediate coupling from its weak-coupling kinsman.

The paper is organized as follows: In Section 2 we introduce the timescales, $t_m$ and $t_{hyb}$,

that characterize the spin and charge dynamics. In Section 3, we then describe the evolution of said timescales in the interaction vs. temperature phase diagram of the Hubbard model. Section 4 discusses microscopic insights gained from the timescales and points out connections with research on specific correlated materials, non-equilibrium physics and quantum-classical hybrid approaches. Finally, we conclude in Section 5 with a perspective on what are the spin-dynamics implications for experimental measurements of magnetic moments.

# 2 Method

## 2.1 Model

We study the simplest model to encapsulate the competition between itineracy through valence fluctuations and local moment physics: the Hubbard model. In the usual notation, its real-space Hamiltonian reads

$$\hat{H} = -2t \sum_{\langle i,j \rangle, \sigma} \hat{c}^{\dagger}_{i\sigma} \hat{c}_{j\sigma} + U \sum_i \hat{n}_{i\uparrow} \hat{n}_{i\downarrow} \tag{1}$$

We consider the half-filled case (one electron per site $i$), with a nearest-neighbor hopping $t$ on the Bethe lattice with infinite coordination (bandwidth $W = 4t$) and suppressed antiferromagnetic order (fully frustrated lattice). The model is solved (numerically) exactly by dynamical mean-field theory [3] (DMFT), for which we use a continuous time quantum Monte Carlo algorithm [11], as implemented in w2dynamics [12]. We measure all energy scales and inverse timescales in units of the hopping $t$. Phase diagram heatmaps are obtained by interpolating results for the discrete set of $(U, T)$-coordinates indicated in the appendix Fig. 11d.

## 2.2 Magnetic susceptibilities

In this setting, we compute the *local* magnetic susceptibility in imaginary time $\tau$ defined as

$$\chi_m(\tau) = g^2 \left\langle \mathcal{T}_\tau \hat{S}_z(\tau) \hat{S}_z(0) \right\rangle \tag{2}$$

where $g = 2$ is the electron's gyromagnetic ratio, $\mathcal{T}_\tau$ is the time-ordering operator, $\hat{S}_z = \frac{1}{2} \left( \hat{n}_\uparrow - \hat{n}_\downarrow \right)$ the $z$-component of the spin-operator of any site $i$. From the response in time, the observable *static* local magnetic susceptibility is obtained through

$$\chi_m(i\omega = 0) = \int_0^\beta d\tau \chi_m(\tau) \tag{3}$$

where $\beta = 1/(k_B T)$. In the following, we will somewhat interchangeably refer to susceptibilities or local moments, as both are linked. In particular, using Curie-like expressions, $\chi_m \propto \mu^2/T$ we will refer to three types of local moments: (1) the instantaneous local moment, $\mu_{inst}$, which derives from the instantaneous susceptibility, $\mu_{inst} = \sqrt{\lim_{\tau \to 0} \chi_m(\tau)} \le 1$ for $S = 1/2$, $g = 2$; (2) the persisting local moment $\mu$ extracted from long (Matsubara) times $\mu = \sqrt{\chi_m(\tau = \beta/2)}$ (see Appendix A); and (3) an effective local moment $\mu_{eff}$ extracted from the static susceptibility via

$$\chi_m(\omega = 0) = \mu_{eff}^2/(k_B T). \tag{4}$$

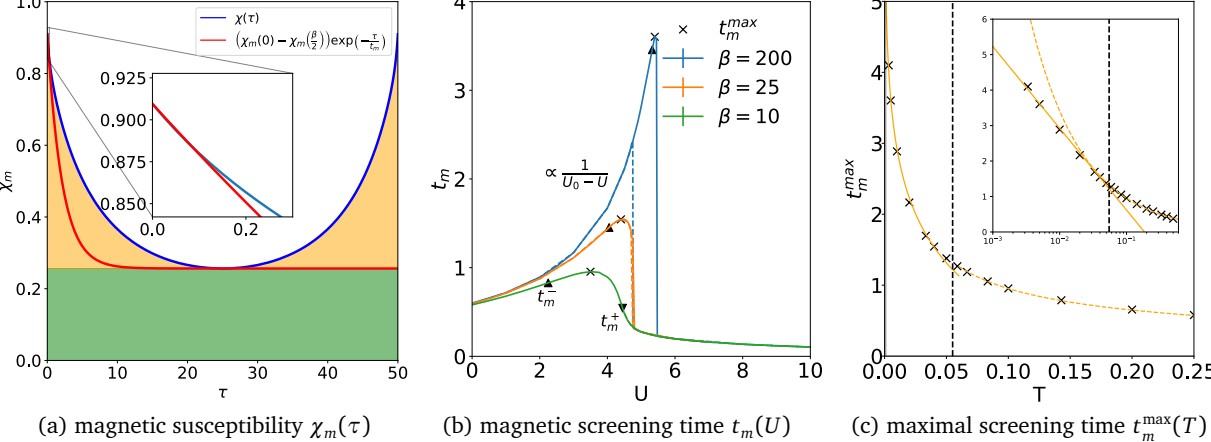

(a) magnetic susceptibility $\chi_m(\tau)$     (b) magnetic screening time $t_m(U)$     (c) maximal screening time $t_m^{\max}(T)$

Figure 2: (a) Local magnetic susceptibility in imaginary time $\chi_m(\tau)$ (blue line); example for $U = 4.9$ and $\beta = 50$. For small times, the susceptibility decreases exponentially with $\tau$. We extract the characteristic timescale $t_m$ of the decay according to Eq. (5) (red line, see also the inset). The static susceptibility $\chi_m(i\omega = 0)$ is the integral over $\chi_m(\tau)$. Following Eq. (7), we distinguish the contribution $\beta\chi_m(\tau = \beta/2)$ (green area), interpreted to arise from long timescales, and the dynamical contribution (orange area). (b) $t_m$ for various $\beta$ obtained for increasing/decreasing $U$ (solid/dashed). Indicated by symbols are the maxima of $t_m(U)$ (crosses) and, above the coexistence region, the inflection points $t_m^-$, $t_m^+$ (up, down triangles). The corresponding $(T, U)$-points of $t_m^{\max}$, $t_m^\pm$ are reported as lines in the phase diagrams Fig. 1 and Fig. 3. Errors of extracted points are always smaller than the symbols. Below the critical U, up to the first inflection point (up-triangles in panel (a)), to excellent approximation: $t_m(U) = -a(U - U_0)^{-1}$ with $a \approx 4$, $U_0 \approx 6.5 \pm 0.1$; above $U_{c2}$, in the Mott insulator: $t_m(U) \propto (U - U_0')^{-\alpha}$ with $\alpha \approx 1/2$, $U_0' \approx 4.0 \pm 0.1$ (c) Maximal timescale $t_m^{\max}(T) = \max_U t_m(T)$ as a function of temperature, as extracted from (b). Above the critical end point, $T_c \approx 0.055$ (dashed vertical line), we find $t_m^{\max} \propto T^{-\alpha}$ (dashed orange line) with $\alpha \approx 0.27$; below $T_c$ the divergence turns logarithmic, $t_m^{\max} \approx -\log(T/\gamma)$ (solid orange line), with $\gamma \approx 0.19$. The inset shows the same data on a logarithmic $T$-axis.

## 2.3 Definition of timescales

### 2.3.1 Local moment screening

Fig. 2(a) displays a representative example of $\chi_m(\tau)$ at intermediate coupling strength. For all interactions $U$, $\chi_m(\tau)$ decays exponentially at short times, $\tau \ll \beta/2$, allowing us to define a characteristic timescale $t_m$ for the screening of the local moment through

$$\chi_m\left(0 \le \tau \ll \frac{\beta}{2}\right) = \chi_m\left(\tau = \frac{\beta}{2}\right) + \tag{5}$$
$$+ \left[\chi_m(\tau = 0) - \chi_m\left(\tau = \frac{\beta}{2}\right)\right]e^{-\tau/t_m}$$

as indicated in Fig. 2(a). The time $t_m$—extracted through a fit for $\tau \in [0 : 1/8]$—is a measure for the speed with which the instantaneous susceptibility, $\chi_m(\tau = 0)$, decays towards its limiting value at $\tau = \beta/2$. Following common practice [13–15], we interpret $\chi_m(\tau = \beta/2)$ as the magnetic susceptibility of the long-time limit (see Ref. [16] for context and Appendix A for a discussion). Then, $\chi_m(\tau = \beta/2)$ signals the presence of a *persisting* local moment, i.e., a fluctuating spin moment that does not decay with time. For an alternative definition of a timescale of magnetic screening, see Appendix B. Analytical results for the screening by non-interacting electrons can be found in Appendix C.

### 2.3.2 Valence fluctuations

For the charge degrees of freedom we introduce a timescale $t_{hyb}$ characteristic for valence fluctuations. In DMFT, the amplitude for the process of an electron visiting a site of the lattice at a time $\tau'$ and to stay until $\tau$, is given by the hybridization function $\Delta(\tau - \tau')$ [2,3]. Therefore, a particularly convenient way to quantify the typical timescale associated with the valence history can be obtained from the low-energy behavior in the frequency domain [17,18] ($\hbar = 1$):

$$t_{hyb} = -1/\text{Im}\Delta(i\nu \to 0) \tag{6}$$

## 3 Results

### 3.1 Evolution of timescales through the phase diagram

We now summarize the trends in $t_m$ as a function of growing interaction $U$ at constant temperature $T$, see Fig. 2(b) for selected cases. Conclusions and microscopic insights will be drawn in the discussion section.

In the absence of interactions, $U = 0$, magnetic screening is controlled by the electrons' kinetic energy $\langle \hat{T} \rangle$: At zero temperature, $t_m = (2|\langle \hat{T} \rangle|)^{-1} = 3\pi/16$ (see Appendix C for a derivation). On the one hand, this direct link quantifies the screening-by-visitation picture mentioned in the introduction: The local moment is dynamically screened by the rate with which electrons hop from site to site. On the other hand, this result validates our definition and extraction of the decay time via Eq. (5).

Moving from weak to intermediate coupling, screening slows down significantly, the more the lower the temperature. For still larger $U$, and for temperatures below the Mott transition's critical endpoint, $t_m$ displays a discontinuous jump at the respective critical interaction ($U_{c2}/U_{c1}$ for increasing/decreasing $U$). The spin screening timescale thus neatly delineates the same coexistence

region (marked in Fig. 1, 3 by blue lines) as found in spectral quantities (e.g., the quasi-particle weight $Z$) or static correlators (e.g., the double occupancy). At strong coupling, beyond the critical interaction, the speed with which the instantaneous moment is screened to its persistent value is fast again. In fact, throughout the Mott phase $t_m$ is shorter than anywhere in the metallic phase.

While the described non-monotonous evolution of $t_m(U)$ is discontinuous for temperature-cuts passing through the coexistence regime (e.g., $\beta = 200$, blue line in Fig. 2(b)), the screening time evolves smoothly in the supercritical region above it (e.g., $\beta = 10$, green line in Fig. 2(b)). Still, the crossover from weak to strong coupling can be characterized by analyzing the structure of $t_m(U)$: We identify a (now broadened) maximum $t_m^{\max}$ in Fig. 2(b), that is reported in Fig. 2(c) as a function of $T$. Further, on either side of $t_m^{\max}$, an inflection point emerges, $t_m^-$ (the loci of $\arg\max_U \partial t_m/\partial U$) for weaker and $t_m^+$ ($\arg\min_U \partial t_m/\partial U$) for stronger coupling. Mapping out these three characteristic $(U, T)$-trajectories, we obtain the orange lines that are superimposed on all phase diagrams, e.g., in Fig. 1 and Fig. 3. These lines will provide new insight into the supercritical Mott crossover as detailed in Section 4.3.

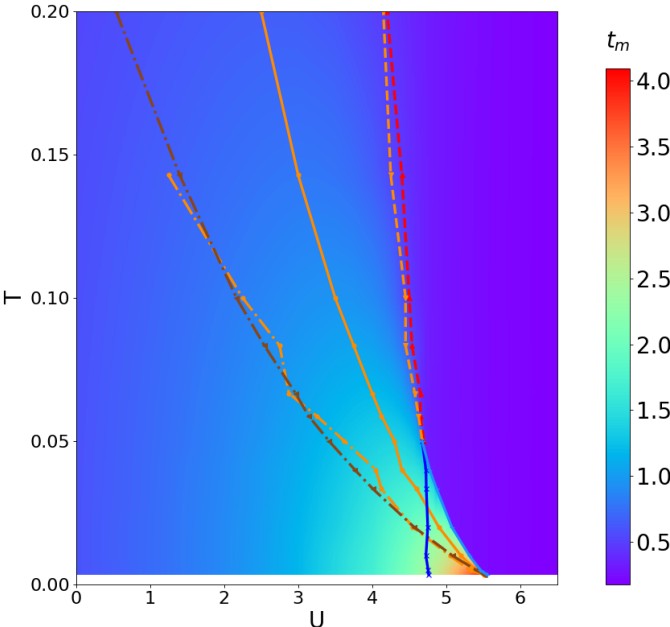

Figure 3: Phase diagram of the Hubbard model characterized by the timescale of magnetic screening $t_m$ (heatmap; see Eq. (5)). Again, orange lines follow the maxima (full) and inflection points (dashed) of $t_m$ at constant $T$. The brown line (dash-dotted) indicates the crossover between a magnetic susceptibility dominated by short-time dynamical contributions (below the line) and a preponderant local moment (above the line). The Widom line is shown in red (dashed). Solid blue lines delimit the metal-insulator coexistence region, inside which we display $t_m$ for the metallic solution.

# 4 Discussion

## 4.1 Timescale behavior

We first discuss the physics at very low temperatures. There, increasing the interaction from weak to intermediate coupling, the local moment survives for longer timescales: $t_m$ becomes larger. This trend is intuitively sound, as the Hubbard $U$ suppresses kinetic energy. However, the rise of $t_m$ with $U$ is faster than $\propto 1/|\langle \hat{T} \rangle|$ suggests (see Fig. 9). A contributing factor is that the unscreened instantaneous moment becomes larger with $U$, as magnetically inactive double occupations, $d = \langle n_\uparrow n_\downarrow \rangle$, and empty sites decrease. Indeed, $\chi_m(\tau = 0) = 1 - 2d = 1/2$ (1) for $U = 0$ ($U = \infty$). At the same time, below the $t_m^{\max}$ line (where $dt_m/dU > 0$), screening is still—as will be discussed below—complete, i.e. $\chi_m(\tau = \beta/2)$ is small. Then, the difference $\chi_m(\tau = 0) - \chi_m(\tau = \beta/2)$, that the exponential decay, Eq. (5), needs to cover, increases with growing interactions (see Fig. 11c), which we find to correlate with longer screening times $t_m$. In the Mott phase, instead, the existence of a persisting local moment renders $\chi_m(\tau = 0) - \chi_m(\beta/2)$ small: screening, while incomplete, is fast.

At the Mott transition, close to $(U_{c2}(T = 0), T = 0)$, many quantities display critical behavior. Indeed, closely approaching the insulator from weak coupling, $m^*/m_0 = Z^{-1} \propto (U_c - U)^{-1}$ for the mass enhancement, $\chi_m(\omega = 0) \propto (U_c - U)^{-1}$ for the static local magnetic susceptibility, and $\langle n_\uparrow n_\downarrow \rangle \approx a + b(U_c - U)/U_c$ for the double occupancy, where $U_c = U_{c2}(T = 0)$ (see Ref. [3] and references therein). Also the decay timescale $t_m$ displays characteristics of critical behavior, see Fig. 2(b). We find $t_m \propto (U - U_0)^{-1}$ for $U$ below the first inflection point $t_m^-$, i.e., $t_m$ approaches a divergence with the same critical exponent as the static susceptibility. This interaction-driven slowing down goes hand-in-hand with a softening of the paramagnon-like low-energy mode in $\chi_m(\omega)$ [19]. However, the Mott transition occurs prior to the divergence of $t_m$: Indeed, $U_{c2}(T = 0) = 5.86$ [20] is notably smaller than the critical interaction $U_0 \approx 6.5$ from the spin dynamics. This finding could be interpreted as the quenching of *charge* fluctuations at the Mott transition to preclude an instability of the *spin* degrees of freedom that would occur at larger interactions. It is then tempting to ask whether obviating the Mott localization—e.g., through light doping—would allow exploring an otherwise hidden spin regime. A likely candidate is a metal phase with frozen spins. Such a phase was previously evidenced in a multi-orbital Hubbard model in an extended region away from half-filling [14], and in simulations for iron pnictides [21] and chalcogenides [22]. In these systems, the Hund's rule coupling $J$ crucially influences the spin [15, 23–25] and charge [26, 27] dynamics. Favoring instantaneous high-spin configurations, the Hund's $J$ (for certain fillings) hinders purely kinetic and Kondo screening [28, 29], while at the same time pushing the critical interaction of the Mott phase to stronger coupling [26]—both promoting a regime with slow spin dynamics. Our results for the one-orbital Hubbard model then may suggest, that frozen spins are more ubiquitous and can appear—at least at low temperatures— even without the spin friction induced by Hundness. Candidate materials where this phenomenon could be observed in an effective one-orbital setting are organic salts [30–34], nickelate superconductors [35], and carbon-group adatoms on semiconductor surfaces [36–39].

To shed more light onto thermal effects in the spin dynamics, we analyze the temperature dependence of the local moment screening. Specifically, we follow the maximal magnetic screening time $t_m^{\max} = \max_U t_m(U)$ as a function of $T$. Upon cooling, $t_m^{\max}$ increases monotonously, see Fig. 2(c), reaching more than 16 times the inverse bandwidth at $\beta = 300$. For temperatures above the critical endpoint of the Mott transition the growth of $t_m^{\max}$ is algebraic: $t_m^{\max} \propto T^{-\alpha}$ with $\alpha \approx 0.27$. Inside the coexistence region, the rise in $t_m^{\max}$ slows down. We find $t_m^{\max} \approx -\log(T/\gamma)$

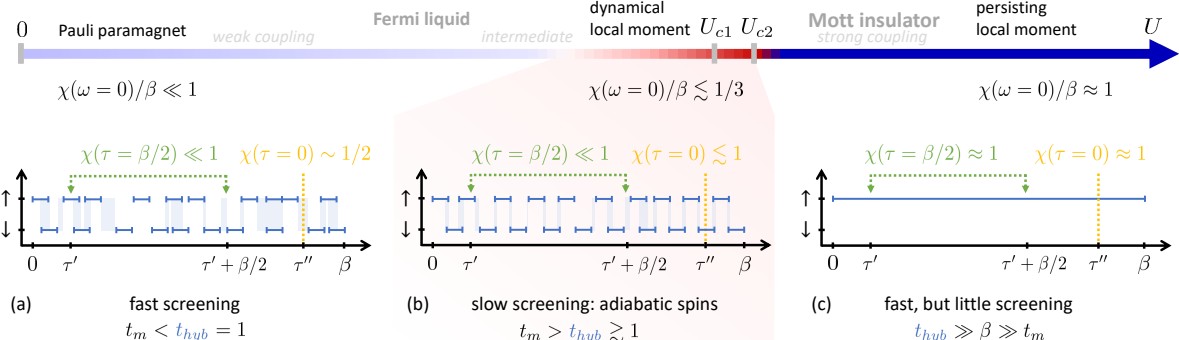

Figure 4: Schematic valence timeline. Shown is a single Monte Carlo configuration in the segment picture for low temperatures (for the metal: below the $t_m^{\max}$ line). Horizontal (blue) lines indicate the occupation of a local site with an $\uparrow$ or $\downarrow$ electron as a function of imaginary time $\tau$; periods with double occupations are highlighted (in light blue). We approximate hybridization events $\Delta(\tau - \tau')$ through the typical timescale $t_{hyb}$ for valence fluctuations of Eq. (6). At low temperatures, $t_{hyb} \approx 1$, in the metallic region while it is exponentially large above $U_{c2}$. Vertical lines (dashed yellow) symbolize measurements of an equal-time correlator $S_z(\tau'')S_z(\tau'')$ that contributes to the instantaneous $\chi_m(\tau = 0)$. Time-delayed measurements $S_z(\tau' + \beta/2)S_z(\tau')$ contributing to $\chi_m(\tau = \beta/2)$ are indicated by (green dashed) arrows. To obtain the susceptibilities, times $(\tau'', \tau')$ need to be integrated and an averaging over many Monte Carlo configurations has to be performed. The intermediate region (shaded in red) is atomic-like at short timescales and Fermi liquid-like at long timescales. We therefore refer to this regime as hosting a *dynamical local moment*.

with $\gamma = 0.19$: The timescale on which the local moment is screened diverges logarithmically towards zero temperature. Accordingly, the spin dynamics freezes at the critical endpoint $U_{c2}(T = 0)$.

It will be interesting to investigate how the slow spin dynamics is affected when non-local exchange interactions are included, either explicitly in the Hamiltonian [40, 41] or effectively, e.g., via cluster extensions [42] to DMFT. On the one hand, exchange can destabilize the persisting local moment in the Mott phase by introducing a spin-liquid-like decaying $\chi(\tau) \propto \tau^{-1}$ [40, 43, 44]—potentially expanding the regime of slow spin dynamics. On the other hand, however, exchange interactions turn the Mott transition first order also at $T = 0$ [40, 42], i.e. $U_{c1} < U_c < U_{c2}$ for the thermodynamic critical interaction $U_c$—presumably cutting-off the $T = 0$ divergence of $t_m^{\max}$.

## 4.2 Adiabatic spin response

To add perspective, we now compare the timescale $t_m$ of the spin degrees of freedom with the time $t_{hyb}$ of Eq. (6) associated with valence fluctuations. In our units, $t_{hyb}$ is pinned to unity throughout most of the metallic regions of the phase diagram (see Appendix C and Fig. 11a). In the Mott insulator, instead, valence fluctuations are quenched and, accordingly, $t_{hyb}$ becomes exponentially large ($\sim \exp(\beta U)$): The Mott localization of charges impedes electrons to hop from site to site. The transition between these two behaviors is abrupt at low temperatures. Only at elevated temperatures, where incoherent charge fluctuations emerge, does $t_{hyb}$ acquire values that are intermediate to one and extremely large.

Combining the individual behaviors of $t_m$ and $t_{hyb}$, we plot in the front Fig. 1 their ratio. In

most parts of the phase diagram $t_m < t_{hyb}$, i.e. the local moment decays faster than the local valence fluctuates (blue regions). However, directly adjacent to the Mott transition, there is a (red) region on the metallic side in which $t_m \gg t_{hyb}$. For the metallic solution shown here, this regime covers most of the coexistence region, extends beyond it down to fairly weak coupling and up to temperatures slightly above the critical endpoint of the Mott transition. The hierarchy $t_m \gg t_{hyb}$ of the characteristic timescales of the spin and charge degrees of freedom implies that, by comparison, the spin dynamics is *adiabatic*.

Such a separation of timescales—a slow magnetic relaxation in a fast electronic medium—is at the heart of approximate schemes, such as the disordered local moment picture [45], Landau-Lifshitz-Gilbert approaches [46], and could guide quantum-classical hybrid theories beyond [47]. Note also that, in this work, we extract the characteristic timescale $t_m$ of magnetic screening from the initial fast decay in $\chi_m(\tau)$. With $t_m$ diverging towards $(U_{c2}(T=0), T=0)$, screening mechanisms active on larger timescales, in particular the Kondo effect, may enter the picture of magnetic relaxation [48, 49].

The separation of timescales in particular provides an intuitive picture to distinguish the Fermi liquid at weak coupling ($t_m < t_{hyb}$) from the Fermi liquid at intermediate coupling ($t_m \gg t_{hyb}$). This distinction has not been observed in spectral quantities [4] nor in static response functions. For instance, an effective exponent in the resistivity, $d \log \rho / d \log T$, provided an intriguing structure around the critical endpoint $(U_c, T_c)$, but, at low temperatures, evaluates to about two—the conventional Fermi liquid value—for any interaction $U < U_{c2}$ [6]. Interestingly, tracing singularities in a more complicated object, the two-particle irreducible vertex function [50–52], provides a horizontal segmentation of the Fermi liquid regime into weak (perturbative) and intermediate (non-perturbative) coupling. The divergencies are linked [53] to ambiguities in the Luttinger-Ward functional for models with Hubbard-like interactions [54] and lead to the breakdown of (bold) diagrammatic expansions [53, 54]. The divergence line at smallest coupling appears closer to the coexistence region (for the Bethe lattice: slightly above $U = 3$ for $T \to 0$ [52]) than our $t_{hyb} = t_m$ criterion and is associated with the density channel [50, 52] and a suppression of the *charge* susceptibility [53]. Our structuring of the Fermi liquid phase, instead, is dominated by the *spin* behavior. In Section 4.1, we even speculated on a decoupled phenomenology, in which the suppression of charge fluctuations at the heart of the Mott transition arrests the crossover into a different spin regime. Non-equilibrium studies could point toward an indirect link of both classifications: It was suggested [50] that the vertex divergence in the density channel separates two different relaxation regimes in the dynamics following an interaction quench [55]. Also, close to the Mott transition a slowdown in the relaxation of degrees of freedom carrying the spin-information was found after ramping-up the hopping amplitude [56]. These empirical links are suggestive of a deeper—yet to be explored—connection between timescales of equilibrium fluctuations and the relaxation dynamics of an out-of-equilibrium state.

We find it instructive to further elucidate the adiabatic spin regime through the time evolution of microscopic quantum Monte Carlo configurations of an individual lattice site: The schematic Fig. 4 displays a single configuration for (a) weak coupling, (b) the adiabatic spin regime at intermediate coupling, and (c) the Mott insulator as a function of imaginary time for low temperatures (for the metal: below the $t_m^{\max}$ line). The presence of an up or down spin is indicated by blue horizontal lines in the top and bottom timeline, respectively, which we approximate to be of typical length $t_{hyb}$. Instantaneous correlators of the given configuration are obtained from measuring the observables at a given time, e.g., $\tau''$ (indicated with a yellow dotted line) and integrating over it. Contributions to retarded correlators with a time-delay $\beta/2$ are also suggested (green dotted arrows).

In the Mott phase, (c) $t_{hyb} \gg \beta$: The valency of a single configuration does not fluctuate in time and hosts a persisting spin moment. The $(\tau'' -)$ *time average* over the equal-time pair of operators $S_z(\tau'')S_z(\tau'') \approx 1/4$ (yellow dashed line) results in a large instantaneous susceptibility $\chi_m(\tau = 0) \approx 1$. Spin symmetry (paramagnetism), is instead recovered through a *configurational average* over many microstates with persisting $\uparrow$ and $\downarrow$ orientation. As indicated in Fig. 4(c) for a measurement with a time delay, $S_z(\tau' + \beta/2)S_z(\tau') \approx 1/4$ (green dashed interval), which results— when averaged over $\tau'$ and configurations—in a persistently large $\chi_m(\tau = \beta/2) \approx 1$. Then, the static susceptibility $\chi_m(\omega = 0)$ is essentially given by $\beta \chi_m(\tau = \beta/2)$—a Curie law. Since $\chi_m(\tau = 0) \approx \chi_m(\tau = \beta/2)$, the persistent moment is reached in a short time: $t_m$ is small (in the schematic Fig. 4(c) it is zero).

Fig. 4(a,b) depict typical spin-configuration histories for the metallic phase, in the sense that we display configurations whose time averages $d = \langle n_\uparrow n_\downarrow \rangle$ and $\langle n \rangle$ are close to the overall configurational averages. Both at (a) weak and (b) intermediate coupling, the timelines basically consists of a random pattern of spin segments of approximate length $t_{hyb} \approx 1$. The only relevant difference between weak and intermediate coupling is the amount that segments of opposite spins typically overlap. These instantaneous double occupations are indicated in light blue. At weak coupling, these instances are frequent, because the system is close to a Slater determinant with equal weights for all states: $[|0\rangle + |\uparrow\rangle + |\downarrow\rangle + |\uparrow\downarrow\rangle]/4$. Averaged over time $(\tau'')$ and configurations, the instantaneous moment is then small, $\chi_m(\tau = 0) \approx 1/2$. At intermediate coupling, in the adiabatic spin regime, double occupations are heavily suppressed. Therefore, integrating the measurement $S_z(\tau'')S_z(\tau'')$ over $\tau''$-time almost picks up the maximum spin moment, $\chi_m(\tau = 0) \lesssim 1$. As to measurements with long time delay, these are small at weak and intermediate coupling. Basically, the probability that consistently a pattern of hybridizations occur that is commensurate between the initial and final time to pick up $S_z(\tau' + \beta/2)S_z(\tau') = (1/2)^2$ is vanishingly small at low temperatures, i.e. $\beta \gg t_{hyb}$.

For long timescales (small energies), the adiabatic spin regime thus shares characteristics with the weak-coupling regime, in that $\chi_m(\tau = \beta/2)$ is small. For short times (large energies), however, the intermediate coupling regime is somewhat atomic like: As in the Mott insulator $\chi_m(\tau = 0) \approx 1$. Note, however, that contrary to the persisting local moment above $U_c$, Fig. 4(c), the large instantaneous local moment in the adiabatic spin regime is *dynamical*, in the sense that for a given configuration, the spin orientation fluctuates in time and both spin orientations contribute to the instantaneous moment. It will be interesting to link this dichotomy of the short and long time behavior of the local moment with the separation of energy scales in spectral properties near $U_c$ [57].

## 4.3 Crossover lines and physical insight

The three characteristic lines extracted from the magnetic timescale—$t_m^-$, $t_m^{max}$, $t_m^+$—structure the phase diagram of the Hubbard model into four regimes, see Fig. 3. We will now characterize these four phases through the lens of the local moment screening. In Fig. 5 we also compare to previous classifications and provide a new perspective on their interpretation. We begin with the weak-coupling phase at low temperatures (left bottom) and work our way clock-wise towards the Mott phase (right).

$t_m^-$: **Emergence of incipient local moments.**    The first inflection point $t_m^-$ in $t_m(U)$ yields the line at lowest temperatures. It starts out at $(U_{c2}(T = 0), T = 0)$, emerges from the coexistence regime by crossing the $U_{c1}$-line at about one third of $T_c$ (the critical end-point of the Mott transition), and

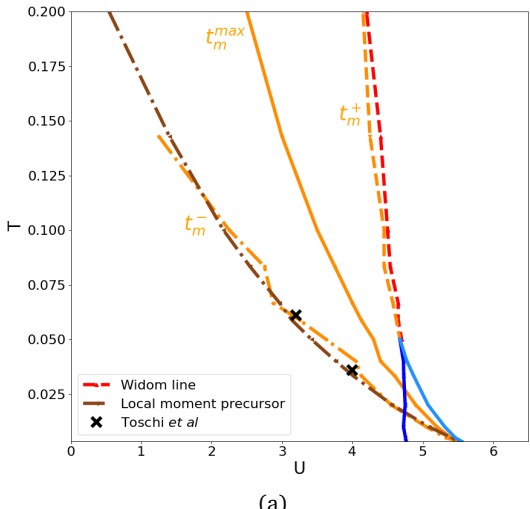
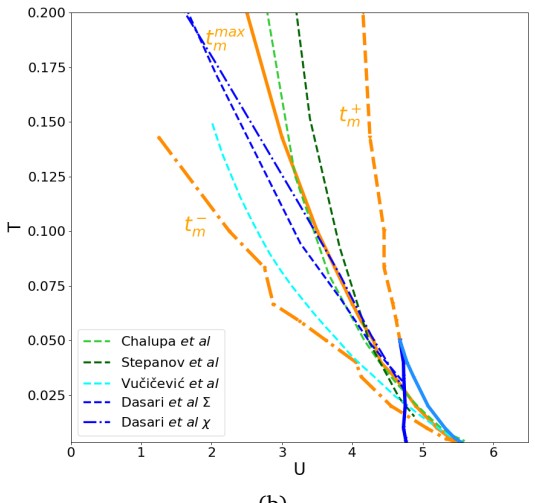

(a)         (b)

Figure 5: Comparison of the timescales (a) $t_m^-$, $t_m^+$ and (b) $t_m^{\max}$ to previous classifications: The $t_m^+$-line (orange dashed) coincides with the Widom line [5–7] (panel (a), dashed red) obtained from $(U_W, T_W > T_c) = \arg\max_U \partial^2 \langle n_\uparrow n_\downarrow \rangle / \partial U^2 |_T$, and marks the crossover from the bad metal phase (left of the line) into the (incoherent) Mott insulator (right of the line). The $t_m^{\max}$-line (solid orange) on which the screening of the local moment is slowest, agrees with various previous crossover lines, see panel (b): Dasari *et al.* [9] found an upper bound of the Fermi liquid regime from the self-energy (blue dashed) and the static magnetic susceptibility (blue dash-dotted), while Vučičević *et al.* [6] extracted a coherence line from the resistivity (cyan dashed). Chalupa *et al.* [58] (light green dashed) and Stepanov *et al.* [59] (dark green dashed, rescaled for the Bethe lattice) proposed lines across which signatures of the local moment emerge. The $t_m^-$-line (orange dash-dotted) coincides with our crossover line (panel (a), brown dash-dotted) on which half of the static susceptibility originates from contributions at long timescales (see Eq. (7)) and is compatible with points extracted from temperature kinks in the electronic specific heat (crosses) found by Toschi *et al.* [10]. The line from Vučičević *et al.* (line "a" in Fig. 4 of Ref. [6]) lies somewhat in between $t_m^-$ and $t_m^{\max}$. We note that (i) this data has been obtain with the less accurate iterated perturbation theory solver; (ii) being based on an *ad hoc* resistivity threshold, there is some freedom in this crossover's precise location. Interpretationwise, this resistivity line describes a coherence-incoherence crossover, i.e. it is closer in spirit to $t_m^{\max}$.

rises into the weak coupling regime. As detailed below, previous criteria to classify the metallic region of the phase diagram yield crossovers at higher temperatures. This in particular implies that both sides of $t_m^-$ are a Fermi liquid, to which it provides additional structure. Anatomizing the spin response, we find that $t_m^-$ marks a crossover *inside* the Fermi liquid phase where the balance between a dominant Pauli susceptibility from itinerant carriers and contributions from an *incipient or preformed persisting local moment* tips in favor of the latter. This inner structure of the spin response cannot be distinguished by looking at the *static* magnetic susceptibility $\chi_m(\omega = 0)$ alone. To gain deeper insight, we scrutinize contributions to $\chi_m(\omega = 0)$ from different timescales.

For this, we heuristically split the static magnetic susceptibility into two terms: [18]

$$\chi_m(\omega = 0) = \int_0^\beta d\tau [\chi_m(\tau) - \chi_m(\beta/2)] + \beta \chi_m(\tau = \beta/2). \tag{7}$$

The first term (orange area in Fig. 2(a)) describes retarded contributions originating mostly from the spin response at short timescales. The second term (green rectangle in Fig. 2(a)) quantifies the part of the susceptibility deriving from the local moment persisting over long timescales. We can then define a line in the phase diagram by requiring both terms to be equal (brown dashed dotted line in all phase diagrams). As seen, e.g., in Fig. 3, this crossover line neatly coincides with $t_m^-$—which can therefore be characterized as a crossover into a Fermi liquid with longer-term spin memory. However, despite the sizable Curie-like $\beta \chi_m(\beta/2)$ contribution to the static susceptibility, the overall effective moment, $\mu_{eff}$ according to Eq. (4), is far from fully formed (see also Ref. [16]). Indeed, $\chi_m(\omega = 0)/\beta$ shown in Fig. 11b only picks up a little when crossing $t_m^-$. Hence, we interpret $t_m^-$ as signaling the emergence of an *incipient persisting local moment*. Given the link between local moment physics and the entropy of the system, a thermodynamic signature of the $t_m^-$ crossover can be expected. And, indeed, we find $t_m^-$ to coincide with the kink in the electronic specific heat [10,60] that separates two linear temperature regimes in $c_V$ of correlated metals (see Fig. 5 (a)). This kink was derived [10] from a detailed knowledge of the electron self-energy that encodes many-body renormalizations for spectral properties [61], in particular, Kondo physics [62]. Congruence with the local moment precursor $t_m^-$ then suggests that thermodynamic properties sense the local lattice sites' magnetic memory even *before* a persisting local moment develops—providing a temperature (vertical) separation of the Fermi liquid phase.

$t_m^{\max}$: **Into the bad metal.** Previous criteria to structure the metallic regime of the Hubbard model were proposed on the basis of Fermi liquid theory: Dasari *et al.* [9] extracted an upper bound for the Fermi liquid regime from the scattering rate (onset of deviations from a self-energy $\mathrm{Im}\Sigma(\omega = 0, T) \propto T^2$) and the static magnetic susceptibility (which is $T$-independent in the Fermi liquid). Vučičević *et al.* [6] analyzed the resistivity of the Hubbard model and proposed various phenomenological criteria to define the crossover into the bad (=non-Fermi liquid) metal. All these lines are consistent with $t_m^{\max}$, the locus in the phase diagram, where magnetic screening is the slowest, see Fig. 5 (b). The qualitative agreement between the above criteria suggests that the timescale $t_m$ is sensitive to the coherence of the metallic state. At $t_m^{\max}$, where the trend in the local moment screening reverses, the effective local moment, Eq. (4), undertakes large steps towards its single ion limit ($\mu_{eff} = 1$). This unscreening of the local moment is signalled by a rising $\chi_m(\tau = \beta/2)$ [63] (see also Fig. 10). Referring to the timeline picture, Fig. 4, the change in the spin degrees can indeed be linked to valence fluctuations becoming incoherent: The average survival time of a valence state (blue lines in Fig. 4) more than doubles, $2 \lesssim t_{hyb} \lesssim 3$ above $t_m^{\max}$ (see Fig. 11a). Accordingly, the probability of picking up the same spin-state after a time-delay increases significantly, and, hence, so does $\chi_m(\tau = \beta/2)$.

This analysis of the emergence of a local moment above $t_m^{\max}$ is consistent with recent works: Chalupa *et al.* [58] interpreted changes in the fermionic Matsubara-frequency structure of the generalized charge susceptibility as hallmarks of the formation of local moments—allowing the extraction of the Kondo temperature (of DMFT's auxiliary Anderson impurity model)[1]. Stepanov *et al.* [59] derived an effective action for the spin dynamics of the (extended) Hubbard model and studied the formation of local moments using Landau's phenomenology. Both, the Kondo

---

[1]Cf. also the very recent work of Mazitov and Katanin [64] for the 2D Hubbard model.

"fingerprint" [58] and the Landau "magnetic moment" [59] lines are in very good agreement with $t_m^{max}$ (see Fig. 5 (b)). This congruence establishes the slope in $t_m$ for varying interaction strength as an easily accessible indicator to distinguish a Fermi liquid from a bad metal.

$t_m^+$**: Into the Mott state.** Next in line is the second inflection point $t_m^+$ of $t_m(U)$. For the metallic solution, it follows $U_{c2}$ inside the coexistence region and emerges from it through the critical endpoint, see Fig. 1 and Fig. 5(a). Lines emanating from the critical endpoint $(U_c, T_c)$ of a first order transition are typically associated with so-called Widom lines [65]: Generally, thermodynamic response functions that evolve discontinuously through a first order transition still exhibit continuous anomalies in the supercritical region from which a Widom line can be constructed, e.g., the maximum in the isobaric specific heat above a liquid-vapor transition. This classical concept has been extended to strongly correlated electrons by Sordi *et al.* [5, 7]. Specifically, at half-filling, a Widom line can be defined through derivatives of the double occupancy $d$ with respect to the interaction at constant temperature. Here, we follow Vučičević *et al.* [6] and determine the Widom line from $(U_W, T_W > T_c) = \arg\max_U \partial^2 d/\partial U^2|_T$.[2] The line thus defined neatly traces the inflection point $t_m^+$, the $(U, T)$-trajectory on which magnetic screening accelerates fastest with the interaction $U$, see Fig. 3.

Widom lines follow the locus of the maximal correlation length [67], powers of which control the scaling region near the (2nd order) critical end point of the underlying transition. Beyond this *spatial* characterization, the Widom line in supercritical fluids was shown to separate regimes of distinct particle dynamics [68]. Our results clearly suggest that this *temporal* distinction extends to the spin dynamics of the supercritical Mott crossover.

## 5 Conclusions & Perspective

In summary, studying the timescale on which local magnetic moments are screened, we provided new insight into the phase diagram of the Hubbard model. The interpretation of previously established phases was refined through the lens of the spin dynamics. Additionally, we identified a new crossover within the Fermi liquid phase from a Pauli-regime into a region with incipient local moments, that we argued to be associated with a previously observed thermodynamic signature in the electronic specific heat [10, 60]. Comparing the timescales of magnetic screening $t_m$ to that of valence fluctuations $t_{hyb}$, we further evidenced a regime in which the spin dynamics is adiabatic $(t_m \gg t_{hyb})$. Indeed, following the line $t_m^{max}$ on which magnetic screening is slowest, the spin dynamics freezes out logarithmically upon approaching the $T = 0$ critical end point of the Mott transition.

The characteristic timescale of the spin dynamics is relevant for the interpretation of magnetic measurements in correlated materials—especially for frustrated systems that avoid long-range magnetic order [31–34]. How a fluctuating local moment manifests in an experiment indeed depends on the typical timescale $t_{exp}$ on which the probe interacts with the system [15, 25, 39]: A "slow" probe $(t_{exp} \gg t_m)$ will dominantly report on the persistent local moment, while a "fast" probe $(t_{exp} \ll t_m)$ may access the instantaneous moment. In that sense, the screening timescale $t_m$ provides a resolution limit for the measurement of a local moment. Equivalently, for inelastic

---

[2]Note that the Widom line is often defined through the inflection point in the double occupancy $d$, $(U_W, T_W > T_c) = \arg\max_U |\partial d/\partial U|$ [7, 66]. This criterion yields a similar line close to the critical endpoint. At higher $T$, however, this line deviates from $t_m^+$ and eventually even crosses $t_m^{max}$.

probes (e.g., inelastic neutron scattering), the inverse of $t_m$ sets the minimal energy cut-off necessary to resolve the instantaneous local moment. Particular care has to be taken if $t_m \sim t_{exp}$: Then, manifestations of external manipulations of the correlation strength—e.g., by pressure or strain [31–34, 69, 70]—are obscured as also the relative degree of temporal averaging changes. Hence, interpreting magnetic measurements requires an intimate knowledge of the timescales of equilibrium fluctuations.

## Acknowledgements

The authors thank P. Chalupa, K. Held, C. Martins, M. Pickem, G. Sangiovanni, E. Stepanov, A. Toschi, and C. Watzenböck for stimulating discussions. LG acknowledges support through the Erasmus programme of the European Union. This work has been supported by the Austrian Science Fund (FWF) through project `LinReTraCe` P 30213. Calculations were partially performed on the Vienna Scientific Cluster (VSC).

## A   Limits to the interpretation of $\chi_m(\beta/2)$ as local moment

In this work, we estimate the amplitude of the persisting local moment $C$ from the susceptibility in the time domain: $C \approx \chi_m(\tau = \beta/2)$. The rationale is that $\tau = \beta/2$ is the largest imaginary time up to which the susceptibility may decay, before the bosonic nature of $\chi_m(\tau)$ demands it to symmetrically rise again. This interpretation of $\chi_m(\tau = \beta/2)$, however, clearly breaks down at weak coupling for high enough temperatures. Indeed, in the absence of interactions, no persisting local moment can emerge. Still, even for $U = 0$ $\chi_m(\tau = \beta/2)$ is finite if the interval $[0 : \beta)$ is short enough (see also Appendix C.2). Indeed, for a Fermi liquid $\chi_m(\beta/2) \propto T^2$. [3, 14] Below, we provide a heuristic argument to quantify the quality of the approximation $C \approx \chi_m(\tau = \beta/2)$.

Watzenböck *et al.* [16] define the persisting local moment $C$ stringently in the frequency domain, through the discontinuous jump of the Matsubara susceptibility with respect to its dc-limit:

$$\chi_m(i\nu_n) = \chi_{reg}(i\nu_n) + C\beta\delta_{n,0} \tag{8}$$

where $\chi_{reg}(i\nu_n)$ is a continuous function, decaying with $\propto 1/i\nu$ for large frequencies. The anomalous part of $\chi_m(i\nu)$ at zero frequency leads to a time-independent contribution to $\chi_m(\tau)$, while $\chi_{reg}(i\nu_n)$ gives a dynamical signal. Therefore, the *curvature* of the (convex) susceptibility at $\tau = \beta/2$, $K = \left(|\partial_\tau^2 \chi_m(\tau)|/[1 + (\partial_\tau \chi_m(\tau))^2]^{3/2}\right)_{\tau=\beta/2}$, indicates how important itinerant contributions to $\chi_m(\tau = \beta/2)$ are. As apparent in Fig. 6 the curvature is small (blue) in parts of the phase diagram relevant to your study (intermediate to strong coupling, temperatures $T < 0.15$). In particular near the coexistence region, inelastic contributions ($\nu_n > 0$) to $\chi_m(\tau = \beta/2)$ are very small, justifying $C \approx \chi_m(\tau = \beta/2)$ for our purposes. Note that, quite intuitively, the locus of maximal curvature at constant $T$ traces $t_m^{\max}$, the line along which the local moment screening is slowest.

With the limitations of the $\chi(\tau = \beta/2)$ proxy, there is one interesting question that we cannot definitively answer: Does the incipient local moment line (brown dash-dotted, see Eq. (7)) asymptotically approach the $U = 0$ axis or does it have an endpoint at finite coupling and temperature? Watzenböck *et al.*'s insight [16] on $C$ will allow pinpointing the precise location of this crossover's endpoint.

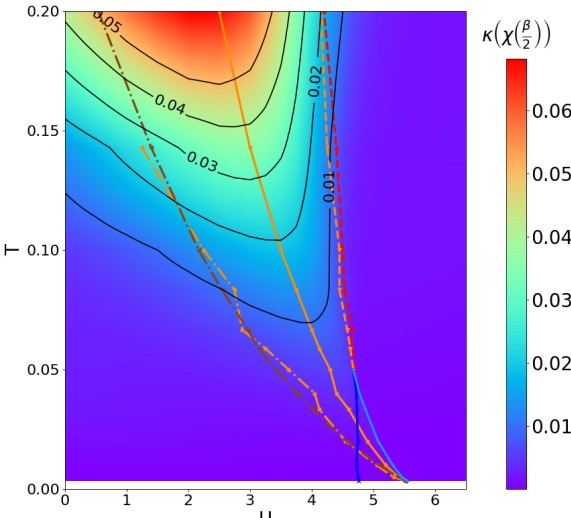

Figure 6: Colormap of the curvature $K$ of $\chi_m(\tau)$ at $\tau = \beta/2$. The curvature quantifies the validity of interpreting $\chi_m(\tau = \beta/2)$ as a signature for a persisting local moment. According to Appendix A, this interpretation holds for a flat susceptibility (blue region).

## B   Alternative definition of the magnetic timescale

Alternative to the magnetic timescale $t_m$ from Eq. (5), one can define a timescale $t_m^0$ via

$$e^{-\tau/t_m^0} = \chi_m\left(\tau \ll \frac{\beta}{2}\right) / \chi_m(\tau = 0) \tag{9}$$

as indicated in Fig. 7. While $t_m$ was a measure for how fast the instantaneous moment decays towards the persisting local moment, $t_m^0$ quantifies the time needed to fully screen the spin moment. If, however, a persisting moment exists, i.e., screening is incomplete, the ansatz Eq. (9) is unphysical and $t_m^0$ accounts for it by drastically increasing, see Fig. 9. Indeed, in the atomic limit, necessarily $t_m^0 \longrightarrow \infty$. Below we will discuss differences between the two magnetic timescales, $t_m$ and $t_m^0$, in the non-interacting limit.

## C   Timescales in the non-interacting limit

### C.1   $T = 0$

In the non-interacting limit ($U = 0$), a closed-form expression can be obtained for the zero-temperature ($T = 0$) local magnetic susceptibility $\chi_m(\tau)$ on the Bethe lattice with infinite co-ordination at half-filling ($n = 1$). Using the density of states of the latter,

$$D(\epsilon) = \frac{\sqrt{4t^2 - \epsilon^2}}{2\pi t^2}, \tag{10}$$

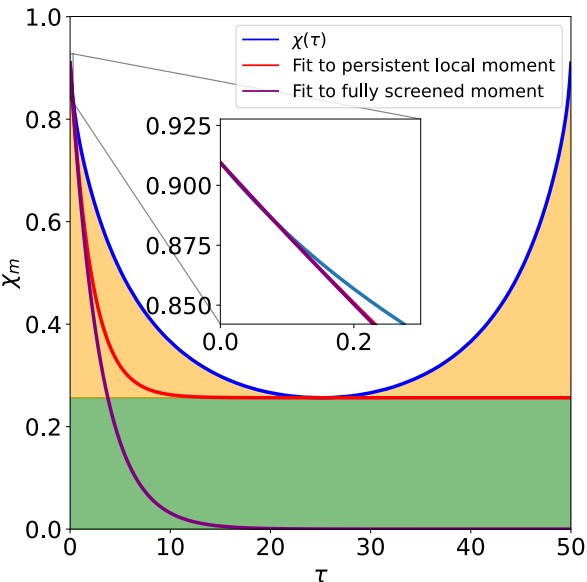

Figure 7: Local magnetic susceptibility in imaginary time $\chi_m(\tau)$ (blue line); example for $U = 4.9$ and $\beta = 50$. Instead of defining the screening time $t_m$ associated with the reaching of a persisting moment, finite offset at $\tau = \beta/2$ (red line), we can extract, via Eq. (9), a decay time $t_m^0$ that assumes the susceptibility to vanish for long timescales (bordeaux line). To account for the persisting local moment, $t_m^0$ has to diverge in the atomic limit.

the local Green's function is, in this case, given by

$$G_0(\tau < 0) \;=\; -\int_{-2t}^{0} d\epsilon D(\epsilon) e^{-\epsilon\tau} \tag{11}$$

$$\;=\; -\frac{I_1(2t\tau) - L_1(2t\tau)}{2t\tau} \tag{12}$$

where $I_n(z)$ and $L_n(z)$ are the modified Struve and modified Bessel functions, respectively. The local magnetic susceptibility then becomes

$$\chi_m^0(\tau) \;=\; \frac{g^2}{2} G_0(\tau) G_0(-\tau) \tag{13}$$

$$\;=\; \frac{(I_1(2t\tau) - L_1(2t\tau))^2}{2t^2\tau^2} \tag{14}$$

where we used a gyromagnetic ratio $g = 2$. Since $\lim_{\tau \to \infty} \chi_m^0(\tau) = 0$, the timescales $t_m$ (Eq. (5)) and $t_m^0$ (Eq. (9)) are identical here. Following the definition, Eq. (5), an analytical expression for the decay with time can be obtained by comparing the short-time expansion of Eq. (14)

$$\chi_m^0(\tau) \;=\; \frac{1}{2} - \frac{8t\tau}{3\pi} + \frac{\left(64 + 9\pi^2\right)t^2\tau^2}{18\pi^2}$$
$$-\frac{92t^3\tau^3}{45\pi} + O\left(\tau^4\right) \tag{15}$$

to the exponential

$$\frac{1}{2}e^{-\tau/t_m} \quad = \quad \frac{1}{2} - \frac{\tau}{2t_m} + \frac{\tau^2}{4t_m^2} - \frac{\tau^3}{12t_m^3} + O\left(\tau^4\right) \tag{16}$$

Here, the factor $1/2$ takes into account that, for $U = 0$, non-magnetic states (empty and doubly occupied sites) make up half of the eigenstate. Equating the linear terms of Eq. (15) and Eq. (16), we obtain

$$t_m = t_m^0 = \frac{3\pi}{16t} \tag{17}$$

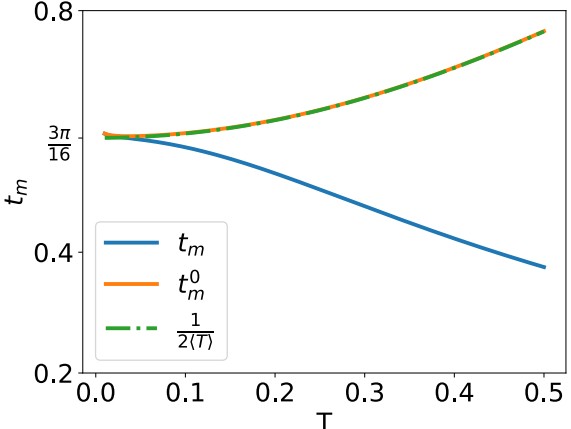

Figure 8: Timescales $t_m$ and $t_m^0$ and half of the inverse kinetic energy, $1/(2|\langle\hat{T}\rangle|)$ as a function of $T$ in the non-interacting limit ($U = 0$): In the absence of interactions, the kinetic energy neatly reproduces $t_m^0$, validating the extraction procedure from an exponential decay.

In the non-interacting case, the screening of the instantaneous moment can only be mediated by the electrons' bare hopping. Computing

$$2\left|\langle\hat{T}\rangle\right| = 2\sum_\sigma \left|\int_{-2t}^0 d\epsilon D(\epsilon)\epsilon\right| = \frac{16t}{3\pi} = 1/t_m \tag{18}$$

we find that the screening timescale equals the inverse of twice the absolute value of the electrons' kinetic energy $\langle T \rangle$. The factor of two is suggestive of the fact that screening-by-visitation is mediated by electrons and holes.

Also the timescale $t_{hyb}$ of valence fluctuations, Eq. (6), is easily derived in the non-interacting case: On the Bethe lattice with infinite coordination the DMFT hybridization function is proportional to the local Green's function, $\Delta(\omega) = t^2 G(\omega)$. From this we directly infer, that for $U = 0$, $\mathrm{Im}\Delta(\omega = 0) = t^2\pi D(0) = 1$ with the density of states $D$ from Eq. (10). Even for finite $U$, the value of the spectral function $-1/\pi\mathrm{Im}G(\omega = 0)$ remains pinned to $D(\epsilon = 0)$, provided that $\mathrm{Im}\Sigma(\omega = 0)$ is negligible [71]. Therefore, $t_{hyb}$ varies little when the system realizes a coherent metal, see Fig. 11a.

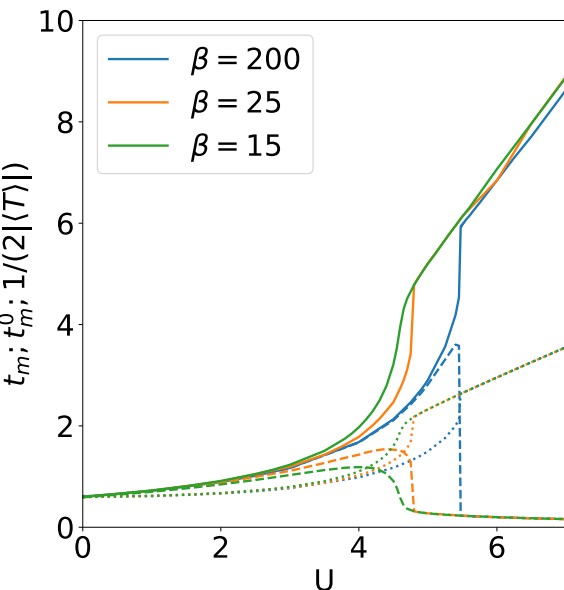

Figure 9: $t_m$ (dashed lines), $t_m^0$ (solid lines), $1/(2|\langle \hat{T} \rangle|)$ (dotted lines) for various values of $\beta$ obtained for increasing values of U. Both $t_m$ and $t_m^0$ grow faster with $U$ than the inverse of the kinetic energy. Above the critical $U$, $t_m^0$ increases significantly to encode the non-vanishing persistent local moment. For low temperatures, all lines merge into the same point at $U = 0$.

### C.2  $T > 0$

At finite temperature, we evaluate Eq. (14) numerically, using the local Green's function

$$G_0(\tau > 0) = -\int_{-2t}^{2t} d\epsilon D(\epsilon) f(\epsilon) e^{-\epsilon(\tau-\beta)}. \tag{19}$$

The kinetic energy is obtained via

$$\langle T \rangle = 2 \int_{-2t}^{2t} d\epsilon D(\epsilon) f(\epsilon) \epsilon. \tag{20}$$

Fig. 8 displays the temperature dependence of the timescales $t_m$ and $t_m^0$. In the limit $T \to 0$ we recover the analytical result Eq. (17). For rising temperature, $t_m^0$ continues to coincide with $1/(2|\langle \hat{T} \rangle|)$ and grows slightly, as expected. The neat agreement between the two quantities in particular validates our fitting procedure for the extraction of characteristic timescales. The timescale $t_m$ instead decreases with temperature, as a consequence of the appearance of a finite $\chi_0(\tau = \beta/2)$. As discussed in Appendix A, in this case $\chi_0(\tau = \beta/2)$ does not signal the presence of a local moment.

# D   The interacting case: $U > 0$

## D.1   Kinetic energy

For the interacting case, we evaluate the kinetic energy as follows:

$$\langle T \rangle = 2 \int_{-\infty}^{+\infty} d\epsilon D(\epsilon) \epsilon \frac{1}{\beta} \sum_n G(\epsilon, i\omega_n) \tag{21}$$

with the lattice Greens function $G(\epsilon, i\omega_n) = [i\omega_n - \epsilon - \Sigma(\omega)]^{-1}$, where $\Sigma(\omega)$ is the DMFT self-energy.

## D.2   Evolution of $\chi(\tau)$ with the interaction $U$

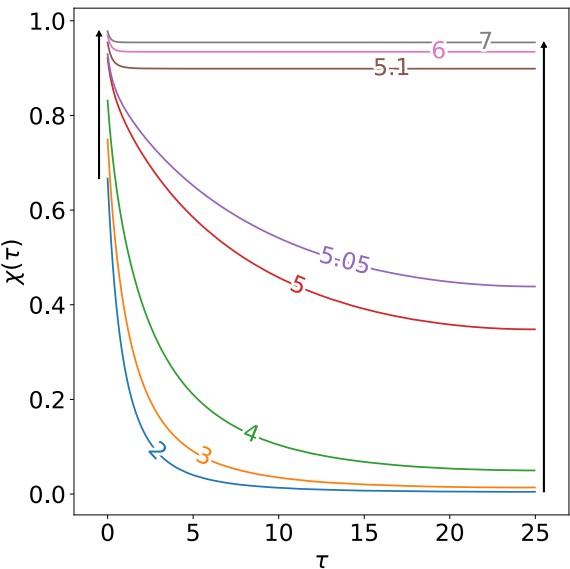

Figure 10: Magnetic susceptibility $\chi(\tau)$ for increasing values of $U$ (specified inline) plotted between $\tau = 0$ and $\tau = \beta/2$ for $\beta = 50$ (i.e. below the critical end-point). The two black arrows indicate the tendency of $\chi(\tau = 0)$ and $\chi(\tau = \beta/2)$ with $U$. At the phase transition (here between $U = 5.05$ and $U = 5.1$), one notices a jump of $\chi(\tau = \beta/2)$ and a sudden change in the global shape of $\chi(\tau)$.

# E   Auxiliary phase diagram heatmaps

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
