# Peer review of "Timescale of local moment screening across and above the Mott transition"

_SciPost Physics_

## Round 1 · Referee Report · Anonymous (Referee 1) · 2022-1-5

Strengths

1- Paper thoroughly connects with literature - the results complement the previous understanding of the different regimes and crossovers in the phase diagram of the half-filled Hubbard model on the infinitely-dimensional Bethe lattice

2- Results are detailed - phase diagram scans are performed thoroughly and with high resolution

Weaknesses

1- the text is hard to read and follow - there is a lot of vague language, jargon, hand-waving arguments and intuitive explanations, and little formalism. There are also a lot of digressions - the text could be more streamlined.

2- the central quantity (characteristic timescale for magnetic screening, $t_m$) is well defined in terms of the way it is calculated, but its physical interpretation is less clear, despite a big part of the text being devoted to that. Connection to very recent results in Ref. 16 is not well established.

Report

Warnings issued while processing user-supplied markup:

  • Inconsistency: plain/Markdown and reStructuredText syntaxes are mixed. Markdown will be used.
    Add "#coerce:reST" or "#coerce:plain" as the first line of your text to force reStructuredText or no markup.
    You may also contact the helpdesk if the formatting is incorrect and you are unable to edit your text.

The authors calculate the local dynamical spin susceptibility in the DMFT solution for the $d=\infty$ Bethe lattice Hubbard model throughout the half-filled phase diagram. They define two characteristic times - $t_{hyb}$ the time electron typically spends on a single site and $t_m$ the magnetic screening time. They argue that the values of these quantities and their ratio provide a clear way of distinguishing between the different regimes on the phase diagram. The main result of the paper is that close to the Mott transition, at low temperature on the metallic side (red region on Fig.1), there is a regime where $t_m<t_{hyb}$, which depicts a system of well defined, yet oscillating local magnetic moments (contrary to the weak-coupling regime, where a good portion of time the sites are either doubly occupied or empty, thus having $S_z = 0$). This is interpreted as a precursor to Mott transition.

The main issue I find with the paper is that the $t_m$ quantity is not defined precisely enough, and this is mainly for two reasons: 1) the authors do not strictly define their terminology, what they mean by "characteristic time for magnetic screening". In Section 2.3.1 they give the definition in terms of $\chi(\tau)$, but they do not specify the physical meaning. This is the central point in the manuscript where this should have been clarified.

Even the definition in terms of $\chi(\tau)$ is problematic. The authors state that $\chi(\tau)$ decays exponentially at small $\tau$, and they support that with Fig.2a - a zoom in at small $\tau$ comparing the data to an exponential function fit. At that level of zoom, both the data and the exponential function appear linear, as would any other analytic function. Clearly, the data does not follow an exponential law. As far as I understand, an exponential decay signals a spectral gap, but this is not at all a general feature of $\chi(\tau)$. It is not clear to me what information can be extracted from this fit.

2) the meaning of $t_m$ is not discussed in terms of the real-time spin-spin correlation function. A simple example of what $t_m$ represents in an analytical expression for the retarded $\chi^R(t)$ would have sufficed to remove all uncertainty of the reader about the meaning of $t_m$. For example, we could naively assume $\chi^R(t) \sim e^{-t/t_m}\cos(t/T_m+c) $ or something like that. I would expect that there are two characteristic times which might be of interest to look at: the correlation time $t_m$, and the period of oscillation $T_m$. Is this the proposed meaning of $t_m$? Can the authors work out one simple analytical example where their imaginary-time-based definition of $t_m$ is connected to real-time behavior? (After all, $t_m$ is defined as a real time.)

The additional problem is the interpretation of $\chi(\tau=\beta/2)$. In Section 2.3.1 it is stated that (a nonzero?) $\chi(\tau=\beta/2)$ "signals" a "fluctuating spin moment that does not decay with time". It is unclear what is meant. One expects that spins cannot be correlated over infinite amount of time in a disordered phase. It is a very special case of degenerate many-body spectra that leads to such anomalous behavior. The authors only cite in passing the yet unpublished Ref.16 where it is the central result that in the Mott insulator phase $\langle S_z(t\rightarrow \infty) S_z\rangle \neq \langle S_z \rangle\langle S_z \rangle =0$. The point in Ref.16 is that $\chi(\tau=\beta/2)$ is never zero, and that it is not straightforward to extract from it the anomalous term $C$ which yields the persistent correlation. The authors of the present manuscript discuss this only in Appendix A, and the explanations given are insufficient. It appears equating $C$ with $\chi(\tau=\beta/2)$ is an ad hoc hypothesis. The results for $C$ obtained by a more rigorous method are available in Ref. 16. It is unclear why the results are not compared directly to this data to support the hypothesis. However, $\chi(\tau=\beta/2)$ results are not essential for the main findings of the study and are not featured on any of the figures in the main text.

Even if the physical meaning of $t_m$ is clarified, I struggle to understand the significance of identifying the $t_m/t_{hyb}<1$ regime. It is undoubtebly an interesting contribution to the variety of regimes that are found in the Hubbard model phase diagram. Yet, the physical picture that the authors show in Fig.4 has been understood for a long time. Clearly, as double occupation goes down, the $\langle S_z^2 \rangle$ is going to grow, and the spins will be correlated over longer times. The fluctuation frequency will greatly change at the transition. (The new information about the anomalously persisting correlations in the Mott insulator is not in any way contained in Fig.4; large $\chi(\tau=\beta/2)$ in itself does not indicate $persisting$ correlation.) The authors have not given a convincing argument that $t_{hyb} =t_m$ represents a special moment in the evolution of the system towards the Mott insulator. On the other hand, the fact that $t^{max}_m$, $t_m^+$ and $t_m^-$ lines coincide with other crossover lines is expected, and hardly revealing of any new physics.

I cannot see that this work satisfies any of the criteria (Expectations) for the publication in SciPost Physics. If sufficiently improved, the paper should be suitable for a more specialized journal.

Requested changes

1- define propely $t_m$ in terms of real-time spin-spin correlation function, and check that it coincides with the definition in terms of the imaginary-time correlation function

2- give better explanation of the connection between $\chi(\tau=\beta/2)$ value and the anomalous persistent correlation or test this hypothesis in a more explicit way

3- connect with the formalism presented in Ref.16

4- in all discussions make a clear distinction between long lasting (yet exponentially decaying) correlations, and truly persistent anomalous correlations (decaying towards a finite value at $t\rightarrow \infty$). If the claim is that the results reveal persistent correlations, this needs to be thoroughly explained.

5- improve text

6- "In DMFT, the amplitude for the process of an electron visiting a site of the lattice at atime τ′ and to stay until τ, is given by the hybridization function ∆(τ−τ′)" - as far as I understand, it is the other way around: it is the amplitude of the electron leaving a site at a time $\tau'$ and returning at a time $\tau$. This should be clear from the corresponding term in impurity action $\bar{c}(\tau) \Delta(\tau-\tau') c(\tau')$. This physical meaning should be more consistent with the proposed definition Eq. 6.

7- Fig.4 caption: "periods with double occupations are highlighted" - clearly, empty site periods are also highlighted

---

## Round 1 · Referee Report · Anonymous (Referee 2) · 2022-1-6

Strengths

The paper by L. Gaspard and J. M. Tomczak studies the timescale(s) of the local moment in the vicinity of Mott transition. Understanding of what determines these timescales is important for both, understanding the nature and physical properties of local moments, as well as general structure of phase diagram near Mott transition.

Weaknesses

Some statements of the paper require clarification, see below.

Report

The subject of the paper is interesting, as well as some of its results, but in my opinion clearer statements on the results of the paper are needed.

Requested changes

  1. The title of the paper claims study of the timescale of local moment screening. In my opinion this is misleading since according to the results of the paper, only t^{max}_m is associated with screening, as identified in Fig. 5b. At the same time, in the title the authors probably mean more general timescale t_m, which according to their Fig. 3 is substantial in the Mott (local moment) regime. So, I think it is better to reformulate the title to reflect (a) presence of multiple timescales (b) their impact on both, local moment itself and its screening.

  2. Regarding the definition of the time t_m in Eq. (5): is there some argument, that (and when) the susceptibility decays exponentially at short times, or this is just an assumption of the authors? This also concerns Eq. (9), as well as Appendix C. In Appendix C the authors claim that they extract t_m for free fermions. But in fact, no exponential decay is present there, and t_m is evaluated from series expansion at tau<<t_m. Is that also the case for the interacting system? If yes, there is no reason to assume an exponential decay of susceptibility. Also Fig. 2a does not show an exponential decay of QMC data. I think that the truly exponential decay may occur only in the presence of wide plateau of chi(tau), reflecting present local moments (still, some more rigorous arguments would be useful). The same concerns statements of Appendix A, which are also probably true only in the presence of the plateau. This question of the presence of local moments should be discussed before introducing the timescales: obviously there can not be a timescale of local moments if the local moments are not present (like in the non-interacting case).

  3. The explanation of Fig. 5 is misleading. The authors probably plot the position of t^+_m, t^-_m and t^{max}_m on U-T phase diagram, but the caption claims "Comparison of the timescales" instead of "Comparison of the U,T position of the timescales". The same applies to the text.

  4. Eq. (7) is just a trivial identity. It may be useful for interpretation of the data, but I think it is incorrect to call it as heuristic splitting. Is Ref. [18] also really needed in this sentence, or it can be shifted to the discussion of its implication for interpretation of the results?

---

## Round 1 · Referee Report · Anonymous (Referee 3) · 2022-1-19

Strengths

1- The motivation of the manuscript is interesting. The manuscript attempts to provide insight onto one of the most important phase diagrams in modern many-body physics, the DMFT solution of the Hubbard model identifying a quantity whose behavior correlates with most of the lines we can draw using more standard "static" observables

2- The manuscript aims at an intuitive interpretation of the numerical results connecting with basic concepts and intuitive arguments

Weaknesses

1- Despite the central role of the quantity t$_m$, the interpretation is at least vague. Of course the mathematical definition is clear, but the theoretical justification for this choice is not discussed and does not seem to be validated by the numerical data.

2- The presentation is lively, but sometimes the language is excessively evocative and weakly informative. The discussion is not completely linear as some results/arguments are anticipated by rather long digressions.

3- The manuscript fails to achieve its goals

Report

The manuscript presents a study of the single-band Hubbard model solved in Dynamical Mean-Field Theory using CTQMC as a solver. The goal of the manuscript is to show that the main features of the phase diagram correlate with the behavior of a spin-related timescale t$_m$ (screening time). The authors definite it from the short-imaginary time $\tau$ behavior of the local spin susceptibility $\chi(\tau)$ which is fitted to an exponential decay and 1) discuss the correspondence between its variations and the lines we can draw on the phase diagram using more standard markers; 2) argue that this quantity can be used to identify new crossovers within the Fermi liquid region.

My main problem with the manuscript regards the definition of t$_m$. First of all it is not clear if the exponential behavior is an assumption (or theoretical expectations) of if they believe that this behavior emerges from the data as some "best fit". Honestly the latter situation seems hardly the case from the data reported in the inset of Fig. 2(a), where the exponential fit seem to hold only up to frequencies of the order of 0.1t where it is hardly distinguishable from a linear function or an other power-law. Also the non-interacting limit presented by the authors in appendix C confirms this point since the analytical expression of t$_m$ comes from the linear term of the expansion.

On the other hand, I do not see why an exponential decay in imaginary time defines a screening time and why this behavior should be limited to small times. The information about timescales is obviously clearer in real-time quantities where one can define timescales associated to oscillations and damping of a given quantity. A similar analysis has been indeed performed in Ref. 25 for a Multiorbital Hubbard model.

So, in my opinion the procedure is not well justified both theoretically (of course the authors can improve this part, which would be welcome), but the exponential fit does not seem to correspond to the data (at least in the example given by the author, which I assume is not their worst example). So it is very hard to understand the meaning of the timescale they extract.

Of course one can argue that a ill-defined quantity has no reason to correlate with the phase diagram. I think that this argument is not solid. Clearly the authors extract some timescale (or inverse of an energy) which reflects some behavior of a very important observable which is clearly expected to evolve significantly where the phase boundaries and the crossovers of the model are found. Moreover, the single-band Hubbard model within DMFT is characterized by the fact that many observables are controlled by the same energy scales, so that many observables are expected to have important variations in correspondence to the important lines of the phase diagram.

I am also a little skeptical about the comparison between the behavior of t$_m$ and the phase diagram lines. Fig. 3 presents the results on a very large temperature range, so that the critical region is basically invisible making it hard to judge the success of t$_m$ to follow the critical region. As a matter of fact the main result is the agreement between the "Widom line" computed from the double occupancy evolution and t$_m^+$ which does seem to me particularly insightful.

Less importantly, I find that the introduction of t$_{hyb}$ is surprisingly short. Why the time of valence fluctuations does not require the evaluation of a response function? On a more technical/presentation note, a reader who is not expert in DMFT has not tool to understand what the hybridization function is. Some words would be absolutely necessary to make the definition accessible beyond the DMFT community.

For these reasons I can not judge the results which are not related to known lines in the phase diagram as the crossover inside the Fermi-liquid region. In this case the poor definition of the marker limits its predictive power in the absence of independent validations. The discussion is not particularly enlightening in this regard, as the arguments given are quite generic.

Finally, I think the authors elude a complete comparison with Refs. [25] and [16].

All these limitations lead me not to recommend publication in Scipost. I think that the manuscript is far from the criteria of this journal. I also think it requires a substantial revision which seriously addresses all the points I raised before being considered by any (more specialistic) journal.

Requested changes

1- Improve the presentation especially in the definition of the key quantity t$_m$ and limiting the abuse of evocative language

2- Motivate the definition of the procedure to extract t$_m$ possibly contrasting with other estimates of characteristic timescales (or inverse energies)

3- Discuss the numerical validation of the procedure

4- Discuss in more details also the timescale t$_{hyb}$

5- Improve the phase diagram 3 in order to better show also the low-temperature data in the critical region (add an inset or another figure)

6- The discussion of the model is not precise. If we are on an infinite coordination ($z \to \infty$) Bethe lattice, the hopping must be scaled as $t = t^*/\sqrt{z}$ where $t^*$ is the quantity setting the bandwidth and appearing in the DOS.

7- The discussion of susceptibilities vs magnetic moments is in my opinion useless and partially confusing and I would eliminate it

---

## Editorial Decision

awaiting_resubmission